# Clinical Stability of Bespoke Snowman Plates for Fixation following Sagittal Split Ramus Osteotomy of the Mandible

**DOI:** 10.3390/bioengineering10080914

**Published:** 2023-08-01

**Authors:** Soo-Hwan Byun, Sang-Yoon Park, Sang-Min Yi, In-Young Park, Sung-Woon On, Chun-Ki Jeong, Jong-Cheol Kim, Byoung-Eun Yang

**Affiliations:** 1Department of Oral and Maxillofacial Surgery, Hallym University Sacred Heart Hospital, Anyang 14066, Republic of Korea; purheit@daum.net (S.-H.B.); psypjy0112@naver.com (S.-Y.P.); queen21c@hallym.or.kr (S.-M.Y.); ddskjc@hanmail.net (J.-C.K.); 2Department of Artificial Intelligence and Robotics in Dentistry, Graduate School of Clinical Dentistry, Hallym University, Chuncheon 24252, Republic of Korea; denti2875@hallym.or.kr (I.-Y.P.); drummer0908@hallym.or.kr (S.-W.O.); 3Institute of Clinical Dentistry, Hallym University, Chuncheon 24252, Republic of Korea; 4Department of Orthodontics, Hallym University Sacred Heart Hospital, Anyang 14066, Republic of Korea; 5Division of Oral and Maxillofacial Surgery, Department of Dentistry, Hallym University Dongtan Sacred Heart Hospital, Hwaseong 18450, Republic of Korea; 6Department of Dental Science & Technology, Shingu College, Seongnam 13174, Republic of Korea; chunkijeong26@gmail.com; 7Mir Dental Hospital, Daegu 41940, Republic of Korea

**Keywords:** orthognathic surgery, bilateral sagittal split ramus osteotomy, patient-specific plates, bespoke Snowman plates, virtual–actual superimposition

## Abstract

Maxillofacial skeletal surgery often involves the use of patient-specific implants. However, errors in obtaining patient data and designing and manufacturing patient-specific plates and guides can occur even with accurate virtual surgery. To address these errors, bespoke Snowman plates were designed to allow movement of the mandible. This study aimed to compare the stability of bespoke four-hole miniplates with that of a bespoke Snowman plate for bilateral sagittal split ramus osteotomy (SSRO), and to present a method to investigate joint cavity changes, as well as superimpose virtual and actual surgical images of the mandible. This retrospective study included 22 patients who met the inclusion criteria and underwent orthognathic surgery at a university hospital between 2015 and 2018. Two groups were formed on the basis of the plates used: a control group with four-hole bespoke plates and a study group with bespoke Snowman plates. Stability was assessed by measuring the condyle–fossa space and superimposing three-dimensional virtual surgery images on postoperative cone-beam computed tomography (CBCT) scans. No significant differences were observed in the condyle–fossa space preoperatively and 1 year postoperatively between the control and study groups. Superimposing virtual surgery and CBCT scans revealed minimal differences in the landmark points, with no variation between groups or timepoints. The use of bespoke Snowman plates for stabilizing the mandible following SSRO exhibited clinical stability and reliability similar to those with bespoke four-hole plates. Additionally, a novel method was introduced to evaluate skeletal stability by separately analyzing the condyle–fossa gap changes and assessing the mandibular position.

## 1. Introduction

Orthognathic surgery (OGS) involves repositioning of the maxilla, mandible, and alveolar bone segments to improve facial balance and occlusion. The goals of this surgery include enhancing musculoskeletal, dentoalveolar, and soft-tissue relationships, and improving mastication, swallowing, breathing, and occlusion [1]. In 1955, the sagittal split ramus osteotomy (SSRO) technique was introduced for the right side of the mandible under local anesthesia in a patient with Class III skeletal malocclusion [2]. Since then, SSRO has become a widely used osteotomy technique for mandibular OGS; it is capable of both mandibular setback and advancement [3]. In 1970, simultaneous bimaxillary surgery was first performed and published [4]. However, surgical correction of maxillary deformities was reported late owing to the lack of a safe and reliable method for cutting, moving, and securely fixing the maxilla in a new position [5]. In recent bimaxillary surgeries, titanium plates and screws have been used to hold facial bones in their new positions, providing stability and fixation. These plates are customized for each patient by the surgeon and are made of biocompatible, lightweight, and strong titanium. They are bent to fit the patient’s bones and remain in place postoperatively, thereby maintaining the new jaw position. Previously, maxillofacial surgeons used wires for fixation and maxillomandibular fixation (MMF) during OGS. The choice of the fixation method is crucial for successful bone healing in OGS, and the use of miniplates has become a routine procedure for maxillary osteotomy. However, the mandible differs from the maxilla in that it contains the temporomandibular joints (TMJ), which are not discrete ball-and-socket joints. The mandibular condyle rotates and glides within the TMJ [6]. Various techniques for bone fixation following SSRO have been documented, with some instances of intraoral vertical ramus osteotomy performed without any fixation [7,8].

Consequently, the stability of the mandible after surgery is more greatly dependent on the method of bone fixation than the stability of the maxilla. Repositioning the condyle after SSRO is critical because it can affect skeletal stability [9]. However, locating the condyles poses a challenge because SSRO divides the mandible into three parts: the distal segment containing the dentoalveolar region and two proximal portions housing the condyles. The distal segment can be controlled with MMF; however, the proximal segments require a physical location before reattachment to the distal segment using osteosynthesis plates. This task requires excellent precision and expertise. Nonetheless, recent advancements in patient-specific plates (PSPs) have improved the accuracy of condyle repositioning [10,11,12].

Two main methods are used for mandibular fixation in OGS. The first method involves rigid fixation using bicortical screws between the proximal and distal bones following SSRO. The second method employes osteosynthetic miniplates with unilateral cortical bone screws, known as “functional stability” [13,14]. Miniplates were first introduced in 1973 and have since undergone various developments for clinical application [15,16]. They are primarily used for segmental interosseous fixation following mandibular osteotomy [17].

The argument for bespoke surgery is based on the fact that surgical procedures are imperfect constructs that apply to diverse patient populations [18]. Computer-assisted surgery (CAS) and PSPs have revolutionized OGS by improving efficiency, accuracy, preoperative planning, and reducing operation time [19]. Recent studies have shown successful positioning of the maxilla using PSPs and CAS, indicating the applicability of bespoke surgery in OGS [20]. However, the application of CAS in mandibular surgery, particularly sagittal split osteotomy, presents more significant differences than actual surgical outcomes [21]. The mandible poses challenges in applying customized surgical guides, and the variability of fracture patterns in the mandible [21] affects the fit of the bespoke plates.

The bespoke Snowman plate was developed to address these challenges and reduce errors during mandibular surgery. This plate design allows for the physiological movement of the mandibular condyle while providing stability. Resembling the shape of a snowman, the Snowman plate features two round holes and a central oval hole. Virtual surgical planning is used to prevent unwanted rotation of the proximal bone fragments. The Snowman plate can act as a conventional sliding plate when the screw is not placed in the anterior hole of a part fixed to the distal segment of the mandible [22,23,24,25,26,27]. Although sliding plates play a crucial role in stabilizing the condyle and allowing proper movement of the proximal segment, further research is warranted to understand the long-term clinical stability of patient-specific sliding plates, including the bespoke Snowman plate used in this study. An accurate assessment of these plates requires a three-dimensional (3D) analysis. Therefore, this study aimed to compare the stability of the mandibular skeleton using bespoke four-hole plates and Snowman plates following SSRO using the 3D method. We hypothesized that there would be no significant difference in the postoperative skeletal changes between the two plate types.

## 2. Materials and Methods

### 2.1. Patients

The study followed the World Medical Association Declaration of Helsinki on Ethics in Medical Research. Approval was obtained from the University Hospital (IRB No. 2018-09-025-001), and informed consent was obtained from all the patients. The patient selection criteria included adult patients who underwent SSRO between September 2015 and July 2018 and who had undergone preoperative orthodontic treatment. The exclusion criteria included patients with cleft palate or craniofacial syndrome, patients in whom the mandibular plate was removed within 1 year of surgery (depending on the circumstances [24]), and patients in whom three screws were used for the Snowman plate application. Figure 1 illustrates the various types of plates used for bone fixation following mandibular surgery. After applying the inclusion and exclusion criteria, 22 patients were included in the study and divided into two groups: the control group (Figure 2), which consisted of 11 patients with bespoke four-hole plates, and the study group (Figure 3), which consisted of 11 patients with bespoke Snowman plates. In both groups, each plate was fixed using four screws. Cone-beam computed tomography (CBCT) scans were performed before surgery (T0), 4 months after surgery (T1), and 1 year after surgery (T2). The virtual surgery image was named T virtual (Tv). T1 was selected as the timepoint based on the reported complete healing of the cortical bone after approximately 16 weeks of fractures [28].

### 2.2. Virtual Surgery (VS), Designing and Creating Patient-Specific Materials, and Actual Surgery

#### 2.2.1. Virtual Surgical Planning (VSP), including Preoperative Preparation and Creating Patient-Specific Surgical Guides and PSPs

The digital workflow of this study followed the sequence shown in Figure 4. Clinical photographs were obtained after patient examination, and a CBCT scan (Alphard 3030, Asahi, Inc., Kyoto, Japan) was performed 2 weeks before surgery to obtain a 3D image. The CBCT images were acquired in the Frankfort plane, parallel to the horizontal plane, using a field of view measuring 200 × 200 mm, a voxel size of 0.39 mm, and exposure conditions of 80 kVp, 5 mA, and 17 s.

A centric relation (CR) wax bite was used to ensure accurate scanning of the condyle in the CR position. However, owing to the bracketing, blurring, and image magnification caused by fixed metal aligners or brackets commonly used in orthodontic patients, the tooth structure could not be precisely obtained through CBCT. Therefore, conventional impression methods were employed, and plaster casts of the upper and lower teeth were created for each patient. These casts were then scanned using a desktop model scanner (Freedom HD; Dof, Inc., Seoul, Republic of Korea) and digitized in the surface tessellation language (STL) format.

Digital Imaging and Communication in Medicine (DICOM) files were obtained from the CBCT results and reconstructed in 3D. The CBCT and plaster model scan files were sent to the digital center, where a digital engineer imported the DICOM and STL files into the planning software to align the patient’s CBCT scan with the dental cast scan image. The semiautomated merging process began by integrating the 3D tooth image from the dental plaster cast into the CBCT image. Manual registration was performed, in which three anatomical measurement points on the dentition were selected to fine-tune the 3D image. The contours of the dental plaster model image were superimposed on the CBCT image, and the necessary adjustments were made.

Next, the skull image was reoriented to match the views of the surgeon, orthodontist, and digital technician, resulting in a final virtual hybrid skull dentition 3D image, also known as a virtual face. The reoriented image was communicated to the surgeon through a computer screen, facilitating discussions with the digital technicians. The surgical plan was determined, including the osteotomy position, movement of the bone segments, PSP position, and final occlusion. After confirmation by the surgeon, patient-specific virtual materials, including osteotomy guides, PSPs, and occlusal appliances, were designed (Figure 5 and Figure 6) (Appendix A). The images of each component were saved as STL files with the surgeon’s approval of the designed materials.

The osteotomy guides and splints (intermediate and final splint) were produced using a 3D printer (HALOT-SKY, Creality 3D, Shenzhen, China) and resin (ODS, Incheon, Republic of Korea). On the other hand, the PSP (patient-specific plate) was created using a five-axis CNC milling machine (ZX-5SM, Manix, Anseong, Republic of Korea) and a medical titanium disc (Grade IV, Arum Dentistry, Daejeon, Republic of Korea).

#### 2.2.2. Surgery

OGS was performed by a skilled surgeon (BEY) using the Obwegeser–Hunsuck method [29] in the mandible. The surgical procedure is described below.

The surgical access for the sagittal split osteotomy was carried out using standard procedures. The surgeon exposed the temporalis muscle insertion, extending it to at least the level of the sigmoid notch, and retracted the soft tissues on the anterior ramus of the mandible. Subsequently, a periosteal elevator was carefully positioned under the periosteum on the inner side of the ramus, above the mandibular foramen. Once the inferior alveolar nerve was identified at the lingula, a nerve hook was inserted into the mandibular foramen and lifted anteromedially.

Next, a patient-specific osteotomy guide, containing information about bone hole placement, was employed. Using a drill, the surgeon created bone holes to accommodate the insertion of screws. The mandible was then horizontally cut using an ultrasonic surgical instrument, effectively dividing it into two separate segments. In accordance with the Hunsuck technique, the medial cut extended to the lingula on the inner aspect of the lingual surface of the ramus (referred to as short-cut medial osteotomy) [30].

Following the osteotomy, a final splint was applied to realign the separated mandibular segments, effectively correcting the jaw deformity. The bespoke plate was aligned with the preformed bone holes, and screws were inserted and fixed. Once the final splint was removed, the movement of the mandible was evaluated. Customized proximal segment positioners were then used to ensure the proper arrangement of the proximal segments (condyle-bearing segment) and distal segment (tooth-bearing segment), matching the planned virtual surgery [31]. The final positions of the proximal and distal segments were cross-verified using both the bespoke plates and the proximal segment positioners to ensure precise alignment with the virtual surgery position. Once it was confirmed that the mandible had been appropriately repositioned, suturing of the mucoperiosteum was performed.

If maxillary surgery was required, conventional Le Fort I osteotomy was performed. During the procedure, the surgeon used a technique known as predictive hole placement [32]. This involved the use of 3D-printed surgical guides to aid in osteotomy and screw hole creation (Appendix A). This significantly enhanced the accuracy and precision of screw placement, resulting in a safer and more successful procedure. The surgeon utilized either four-hole plates or Snowman plates to secure the mandibular segments with four monocortical self-drilling screws (6 mm long). Following surgery, the final wafer remained in place for less than 1 week, supported by rubber elastics, to facilitate proper healing.

### 2.3. Methods

#### 2.3.1. 3D Condyle–Fossa Relationship Analysis

A comprehensive analysis of the condyle–fossa relationship in the TMJ was conducted using 3D imaging. The distances were measured weekly by a single evaluator, and three measurements were performed and averaged. Stratovan Checkpoint software (Stratovan Corporation, Sacramento, CA, USA) was used for the analysis [33]. The software facilitated automatic placement of semi-landmarks on the condylar surface following the initial selection of the condylar region (Figure 7). Subsequently, the examiner refined the points on the condylar surface and automatically placed landmarks on the corresponding fossa surface (Figure 8). On the basis of these identified points, the anterior, superior, and posterior joint spaces (AJS, SJS, and PJS) were measured on a 3D surface, analogous to the conventional two-dimensional (2D) analysis (Figure 9).

#### 2.3.2. Post-Superimposition Analysis of the Mandible

In this study, 3D virtual surgery (Tv) images were compared with postoperative CBCT images (T1 or T2). Bone data from the postoperative CBCT DICOM file were converted to an STL file using R2GATE™ software (MegaGen Implant Co., Ltd., Daegu, Republic of Korea). Geomagic Control X by 3D Systems (3D Systems, Rock Hill, SC, USA) was used to superimpose and compare the virtual surgery and postoperative STL files in 3D. Alignment between the virtual (Tv) and actual surgical images (T1 or T2) was adjusted using the N-point alignment option in the program. Mandibular images were aligned with the vertical cut of the mandible at the PSP application site as reference (Figure 10 and Figure 11). In the color map, adequate alignment is depicted in green, while inadequate alignment is indicated by blue or red. For this study, an acceptable range of surface was defined, allowing for an error of −1 mm to +1 mm.

The results were evaluated by comparing five specified measurement points: the most inferior points of the left and right mental foramen, the root apex of the left and right first molars, and the root apex of the right central incisor. The coordinates of each point were recorded by using a trigonometric system (x, y, and z). The *x*-axis represents the left and right directions, the *y*-axis represents the up and down directions, and the *z*-axis represents the front and back directions. A single evaluator measured each point and distance three times, at 1 week intervals, and the measurements were averaged. ΔT1 (T1 − Tv) and ΔT2 (T2 − Tv) were calculated and compared within each group. These values represent the differences between the actual surgery (T1 or T2) and virtual surgery (Tv) measurements in the respective groups.

### 2.4. Statistical Analysis

The chi-square test was used to assess disparities among categorical variables. To explore variations in the joint cavity changes over time, a repeated-measures analysis of variance (ANOVA) was conducted. Within each group, a paired *t*-test was used to scrutinize the differences between the time points. Multivariate ANOVA was used to examine the differences between the 3D surface and each coordinate plane for each independent variable (group, point, and time). Data were analyzed using the statistical package (SPSS ver. 28.0, IBM Co., Armonk, NY, USA). *p*-values below 0.01 were considered statistically significant.

## 3. Results

### 3.1. Patient Composition

The control group consisted of individuals with a mean age of 23.27 ± 1.18 years and a male-to-female ratio of 4:7. In comparison, the study group had a mean age of 22 ± 1.18 years and a male-to-female ratio of 6:5. The study group included a higher proportion of patients who underwent mandibular surgery alone. Most patients in both groups underwent surgery to correct skeletal class III malocclusion (Table 1).

The demographic characteristics of the participants in the control and study groups are shown in Table 2. There were no statistically significant differences in the distribution of patients between the two groups.

### 3.2. 3D Condyle–Fossa Relationship Analysis Results

Repeated-measures ANOVA was conducted to assess the interaction effects between time (T0 and T2), group, and joint space location on the changes in the distance between the condyle and fossa. The findings revealed that the main effect of time was not statistically significant. Moreover, no significant interaction effect was observed among the independent variables (Table 3). Additionally, no differences were observed in the distance between the times at each joint space location for each group, as shown in Table 4.

### 3.3. Post-Superimposition Analysis Results of the Mandible

Multivariate ANOVA was conducted to examine the main effects of the group, point (five points), and time (ΔT1 and ΔT2) on the dependent variables (X, Y, Z, and surface difference), as well as the interaction effects of each variable. The results demonstrated that the Z-difference varied significantly over time (F = 23.525, *p* < 0.01). Furthermore, significant interaction effects were observed between the point and time for X, Z, and the surface differences (*p* < 0.01), as shown in Table 5.

The main effects of group and time were not statistically significant for any dependent variable (Table 6). However, the interaction effects of point and time were significant for the X, Z, and surface differences, as reported in Table 5. However, no significant differences were observed over time for each group or at any point within the group (Table 7).

Upon analyzing the surface difference of a representative case from the control group (Figure 12), it was noted that there was an average difference of −0.12 mm and −0.08 mm for ΔT2 (T2 − Tv) at the left and right mental foramen locations, respectively. Four months post surgery, ΔT1 (T1 − Tv) indicated a difference of 0.23 mm and 0.17 mm at these locations. For the bone surfaces corresponding to the left molar root, right molar root, and incisor root, ΔT2 measured 0.01 mm, 0.06 mm, and 0.37 mm, respectively. Furthermore, ΔT1 displayed differences of 0 mm, −0.17 mm, and −0.72 mm at these locations. Detailed measurements for the control group can be found in Table 7.

Upon analyzing the surface differences in the representative case of the study group (Figure 13), the average ΔT2 (T2 − Tv) at the left and right mental foramen locations was −0.21 mm and −0.08 mm, respectively. The corresponding ΔT1 (T1 − Tv) values at these locations were 0.23 mm and 0.20 mm. On the bone surfaces corresponding to the left molar root, right molar root, and incisor root, the ΔT2 values were 0.13 mm, 0.05 mm, and 0.33 mm, respectively. The corresponding ΔT1 values at these locations were −0.05 mm, −0.03 mm, and −0.50 mm. The detailed measurements of the study group are shown in Table 7.

## 4. Discussion

In a previous study, we demonstrated the accuracy of OGS using a bespoke plate (PSP) in the maxilla [20]. We further wished to investigate the clinical efficacy of the bespoke plates in the mandible. However, SSRO, commonly used in the mandible, presents challenges for applying surgical guides compared to the maxilla, and the bone fracture pattern is more varied [21,34,35]. We conducted a clinical evaluation to assess the impact of a bespoke Snowman plate designed on the basis of sliding plate studies [22,23,24,25,26,27] on postoperative mandibular positioning. The study consisted of a control group using a bespoke four-hole plate and a study group using a bespoke Snowman plate with a sliding plate effect, with the aim of evaluating the stability of PSPs for up to 1 year after surgery. Our findings did not reveal significant differences in the postoperative skeletal changes between the control and study groups. However, an interaction effect was observed between certain independent variables and changes in the mandibular bone. Although the study demonstrated differences in the patient demographics between the two groups, it did not provide details on individual surgeries. Previous studies have highlighted that greater maxillary or mandibular bone movement results in higher relapse rates following OGS [36,37,38,39]. Given the variation in mandibular bone displacement among patients in our study, an interaction effect could exist. Further investigations are necessary to explore the relationship between bone changes and the amount of bone movement as an independent variable.

The fixation method for OGS is vital for preventing relapse. Joss et al. found that rigid fixation with miniplates was associated with a lower risk of relapse than fixation with cortical screws [40]. Roh et al. reported a preference for miniplates over cortical screws [41]. Surgeon preference influences the choice of the fixation method. Miniplates provide advantages, such as the elimination of transcutaneous puncture, easy removal under local anesthesia, and maintenance of condylar orientation within the fossa [42].

CAS and PSP are commonly employed in OGS [19], with the PSP system proving to be more accurate than intermediate CAD/CAM splints in transferring the virtual plan to the operating room [43,44]. Despite improvements in surgical accuracy achieved through virtual surgery simulation and PSP, intraoperative errors can still occur owing to various factors [45,46]. Although CAS and PSP offer benefits [20,47,48,49], solely relying on a virtual plan can lead to deviations from the original plan, emphasizing the need for techniques that compensate for positioning inaccuracies during surgery. In addition, another study demonstrated that customized osteosynthetic plates for OGS exhibited good accuracy in the maxilla but showed higher deviations in the mandible. This observation suggests that further research required to determine the extent of error that cannot be corrected through postoperative orthodontic treatment [50]. Several studies have consistently reported that achieving the final postoperative position is inherently more difficult in the mandible compared to the maxilla [21,50,51]. Therefore, utilization of bespoke Snowman plates, which allow for semi-rigid fixation after SSRO and can accommodate minor errors, may offer a viable solution to address these issues.

Although titanium is commonly used for bone fixation in facial skeletal surgery, efforts should be made to minimize the overall foreign body volume. Despite its general reputation for biocompatibility [52,53], concerns regarding allergy still persist. Notably, the implementation of Snowman plates in this study reduced the amount of titanium used in comparison to the bespoke four-hole plates or ready-made four-hole plates.

Previous studies on OGS primarily relied on two-dimensional (2D) cephalometric images, which have limitations in accurately representing 3D changes [42,54,55]. In this study, we utilized a 3D program based on CBCT to analyze the stability of the mandibular bone and articular cavity because 3D analysis provides higher accuracy than 2D methods [56,57,58]. We employed the TMJ analysis program, which allowed us to set three articular primitives on the condylar surface. The program’s semiautomatic landmarks accurately represented the condyle and articular surfaces, providing more information and facilitating joint cavity measurements. Our joint cavity analysis revealed no difference between the bespoke Snowman plate group and the bespoke four-hole plate group, which is consistent with previous studies [59,60,61]. The 3D method used in our study, coupled with a long-term follow-up of over 1 year, further enhanced the reliability of our findings.

To track changes in the distal segment of the mandible, we superimposed virtual and actual surgical images based on the vertical site of the mandibular ramus osteotomy (mandibular first and second molar areas). Unlike the conventional method of superimposing the immobile skull base, we intentionally adjusted the mandibular condylar segment during virtual surgery to achieve ideal contact between the proximal and distal segments [31], considering that the condyle–fossa relationship might change. Therefore, we separately investigated the condyle–fossa relationship and examined the changes in the mandible by superimposing images of the virtual surgical mandible and the actual mandible after surgery. Furthermore, in contrast to previous studies [31], we based the center of rotation of the condyle on the midpoint between the center and medial pole of the condyle [62,63]. In both groups, the condyle–fossa relationship remained stable between the preoperative (T0) and 1 year postoperative (T2) timepoints, indicating that it can be considered a method for proximal segment control in virtual surgery.

This study had some limitations. First, the sample size was relatively small, with only 22 enrolled participants, which hindered the generalizability of the findings. Second, the study primarily focused on mandibular setback cases, necessitating further research on advancement cases, potentially requiring plate quantity, shape, and size modifications, as well as an increased number of screws [7]. Third, the investigation solely examined the condyle–fossa relationship without exploring volume changes in the condyle or fossa, warranting additional research. Fourth, only five points in the distal segment were analyzed, indicating the need to investigate changes in the proximal segment. Moreover, this study was conducted at a single institution using a retrospective approach. Lastly, multiple biases are inherent in OGS studies, making a comparison of outcomes across centers challenging owing to varying variables, such as preparation, radiographic quality, surgeon expertise, equipment, and postoperative protocols. More extensive clinical studies with extended follow-up periods, and multicenter prospective designs are imperative for reliable results.

The small number of patients in this retrospective study was a limitation. One reason for the small sample size was the use of strict inclusion and exclusion criteria, which excluded several patients who underwent surgery during the study period. However, the superimposed image analysis used in this study was highly reproducible and provided consistent results even with a small sample size. Condylar position analysis programs display a 3D surface that allows viewing of a large amount of information. However, the reproducibility can be compromised if different examiners use specific landmarks on the 3D surface for analysis. In this study, we used the average of three measurements taken by the same examiner one week apart to increase reliability.

Advancements in digital medicine have led to the increased utilization of 3D diagnostics and PSPs in various medical procedures. These innovations offer several advantages, including the elimination of intraoperative plate adjustments and serving as a reference during the surgical procedure [49,59,64]. PSPs eliminate the need for the time-consuming folding and fitting of plates during surgery because they are prefabricated to match the patient’s anatomy. These plates also contain information on the desired surgical outcome, allowing surgeons to assess the progress of the surgery during the procedure. Virtual surgery enables surgeons to modify the plate’s position according to their preferences, while the surgical guide provides positional information to avoid critical anatomical structures, such as nerves and tooth roots. Moreover, in OGS, symmetrical fixation of plates on both sides is crucial to minimize biomechanical issues in mandibular movement, making PSPs superior to stock plates. However, errors can still occur despite these advancements, and using bespoke Snowman plates may help to compensate for such errors during surgery.

The time required to design, manufacture, and deliver PSPs and guides contributes to the discrepancies between virtual and actual surgeries. Studies have reported turnaround times ranging from 15 to 42 days for PSP production and delivery, depending on company support and the location of the OGS center [45,65,66]. This duration is longer than that of conventional or virtually planned splint-guided OGS. In this study, a different approach was employed, where the time from diagnosis to hospital delivery of bespoke materials was approximately 1 week, aiming to minimize tooth movement during orthodontic treatment and reduce errors associated with customized materials. Notably, the plates were fabricated using a subtractive manufacturing method, which distinguishes this study from previous reports [67]. This study excluded patients who underwent three-screw fixation with Snowman plates, a type of fixation involving two screws in the distal segment and one screw in the large oval hole of the proximal segment. This method is typically used in patients with condylar positioning difficulties or anticipated poor postoperative occlusions. Exclusion was necessary to compare groups with an equal number of screws. Snowman plates are customized; however, using three screws can result in an adjustable [68] or sliding plate effect [22,23,24,25,26,27] during surgery. The long-term success of orthognathic reconstructive surgery depends on the long-term stability of surgical correction [39]. This study demonstrates the stability of PSPs in OGS for up to 1 year and highlights the potential benefits of bespoke Snowman plates. Technology aids surgeons in optimizing operating time and achieving favorable surgical outcomes.

## 5. Conclusions

This study found no difference in the 3D condylar position over time between bespoke Snowman plates and four-hole plates used for bone stabilization after SSRO. In patients with bespoke Snowman plates, the bone surface position changed by no more than 0.5 mm in postoperative images at 4 months and 1 year. This indicates comparable stability between the two plate types. This study proposes a protocol to separately investigate the condyle–fossa space and 3D mandibular changes when assessing mandibular volume changes following OGS.

## Figures and Tables

**Figure 1 bioengineering-10-00914-f001:**
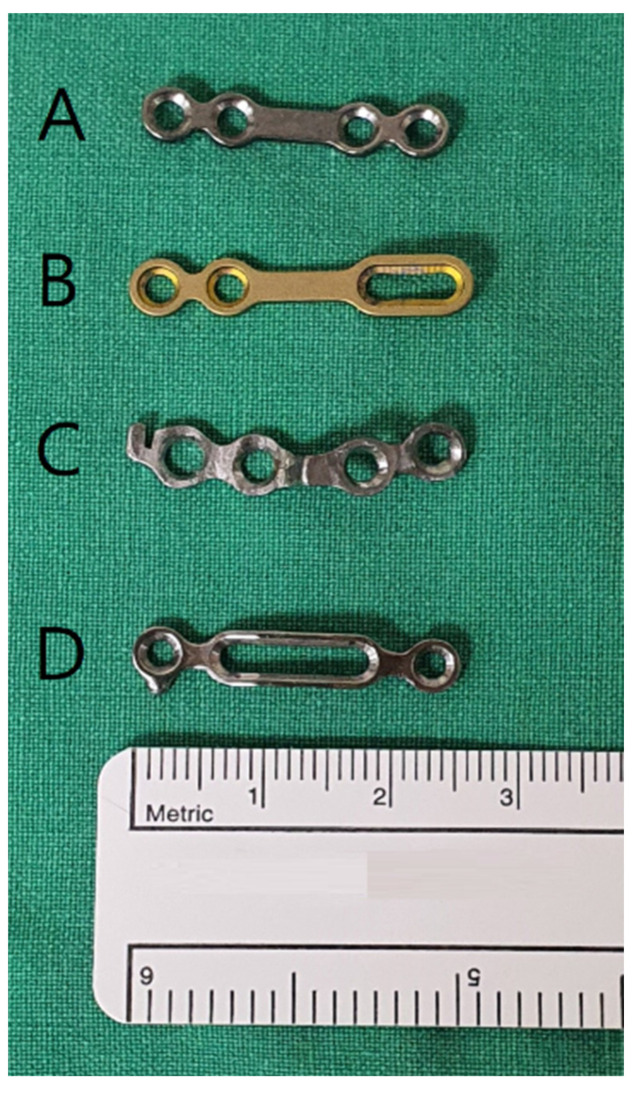
Types of titanium plates used in orthognathic surgery: ready-made four-hole plate (**A**), readymade sliding plate (**B**), bespoke four-hole plate (control group) (**C**), and bespoke Snowman plate (study group) (**D**).

**Figure 2 bioengineering-10-00914-f002:**
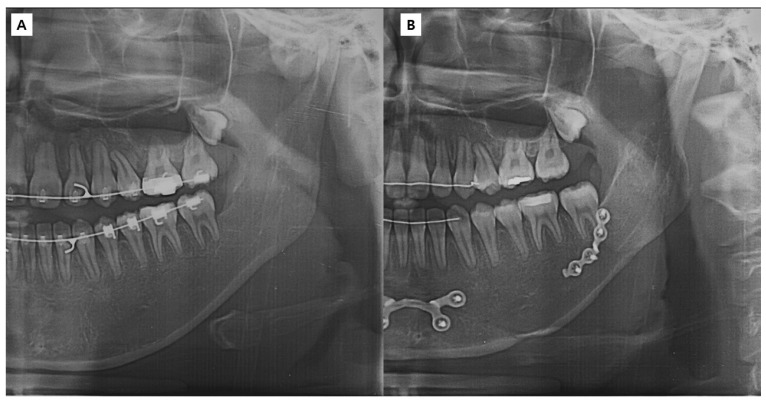
Patient with a bespoke four-hole plate (control group): preoperative panoramic view (**A**) and postoperative 1 year panoramic view (**B**).

**Figure 3 bioengineering-10-00914-f003:**
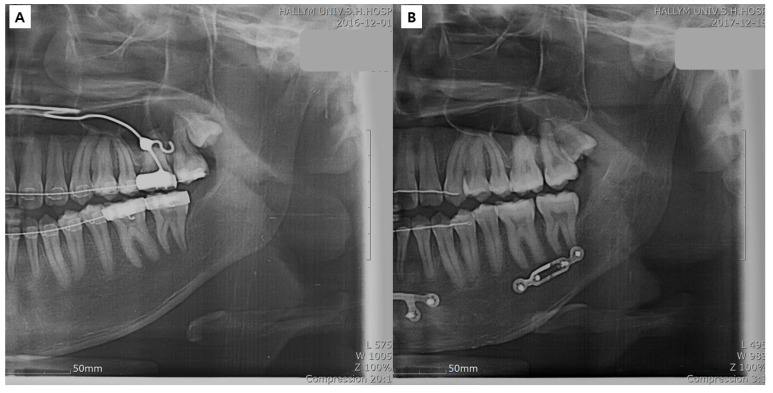
Patient with a bespoke Snowman plate (study group): preoperative panoramic view (**A**) and postoperative 1 year panoramic view (**B**).

**Figure 4 bioengineering-10-00914-f004:**
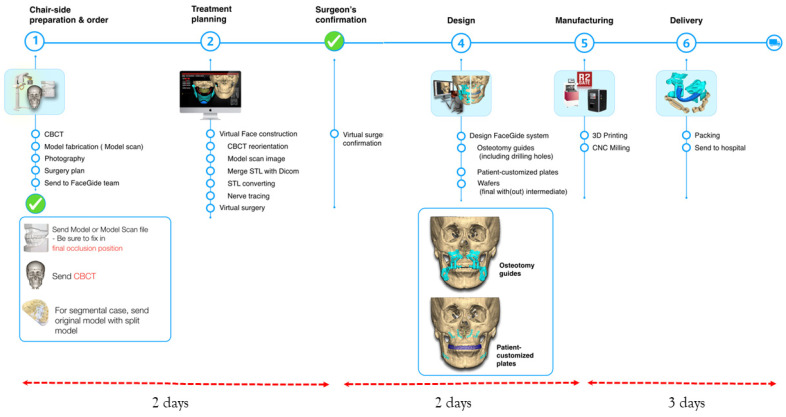
Workflow of the Digital Bespoke orthognathic surgery.

**Figure 5 bioengineering-10-00914-f005:**
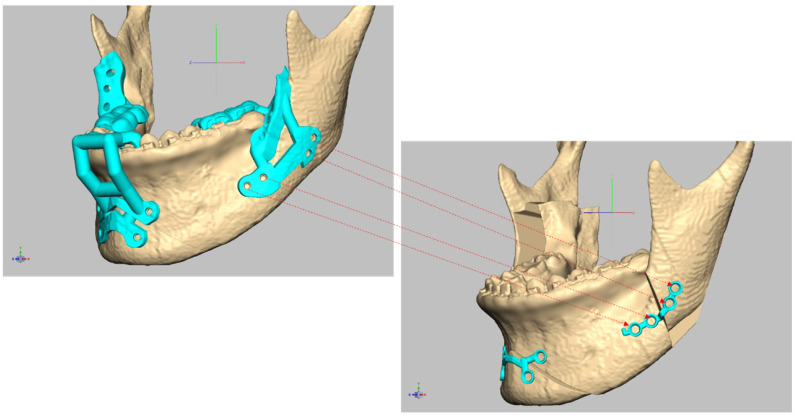
Bespoke surgical guides (**Left**) and bespoke four-hole miniplate (**Right**) (control group) designed for orthognathic surgery.

**Figure 6 bioengineering-10-00914-f006:**
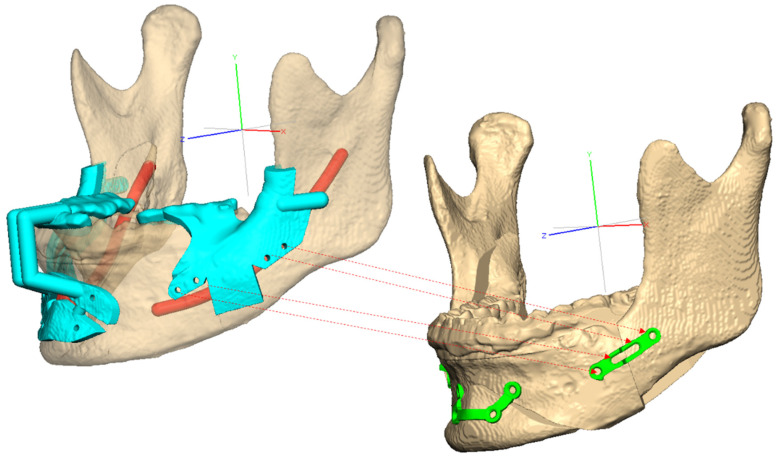
Bespoke surgical guides (**Left**) and bespoke Snowman plate (**Right**) (study group) designed for orthognathic surgery.

**Figure 7 bioengineering-10-00914-f007:**
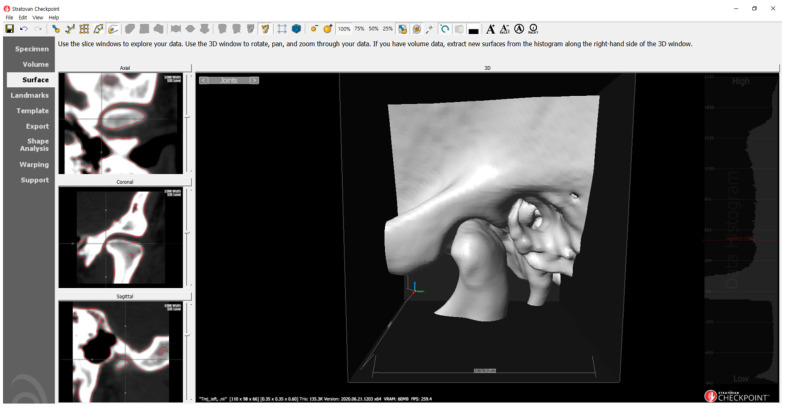
The condylar area is cropped from the patient’s cone-beam computed tomography (CBCT) data.

**Figure 8 bioengineering-10-00914-f008:**
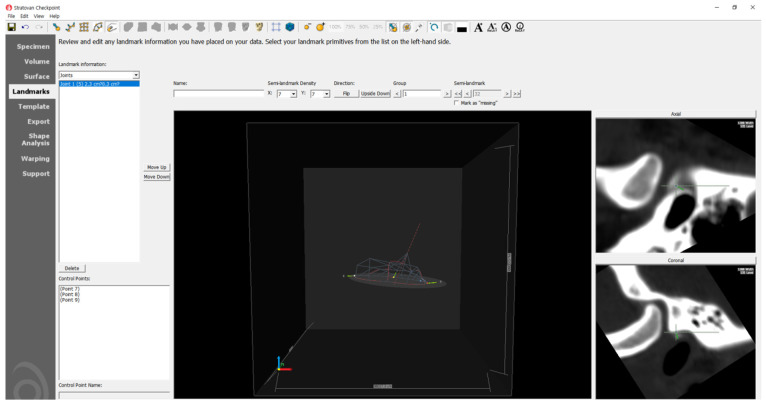
Placing semi-landmarks and adjusting steps.

**Figure 9 bioengineering-10-00914-f009:**
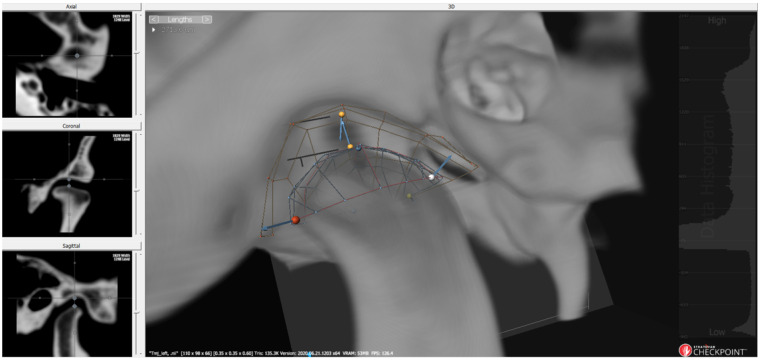
Condyle space measurement in the three-dimensional surface. Two yellow dots present superior joint space.

**Figure 10 bioengineering-10-00914-f010:**
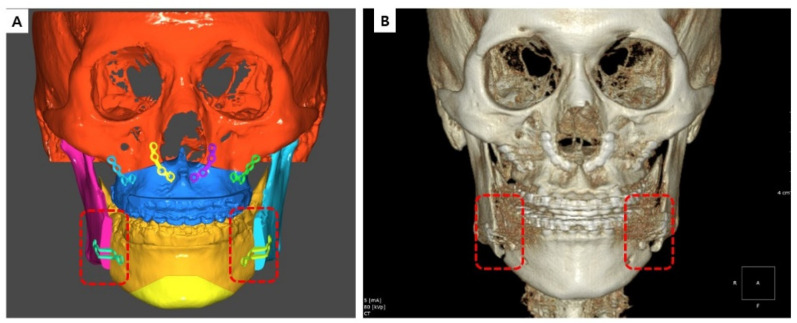
Virtual surgery image (**A**), actual surgery image (**B**), and two images superimposed on the basis of the red square box.

**Figure 11 bioengineering-10-00914-f011:**
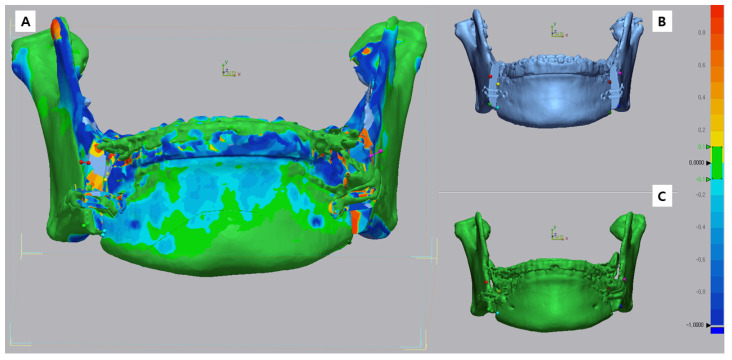
B and C superimposed image (**A**), virtual surgery image front view surface tessellation language (STL) image (**B**), and actual surgery image front view STL image (**C**).

**Figure 12 bioengineering-10-00914-f012:**
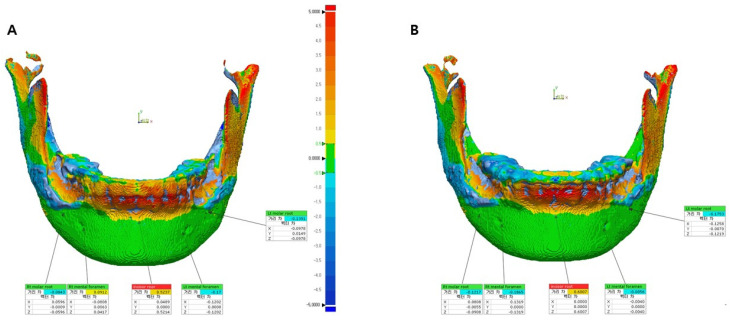
Comparison of five landmarks located on the bone surface after the superimposition of T2 and Tv (ΔT2) (**A**) and T1 and Tv (ΔT1) (**B**) in patient 4 of the control group.

**Figure 13 bioengineering-10-00914-f013:**
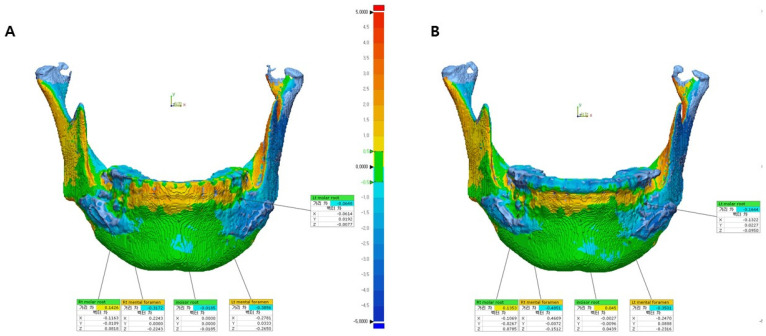
Comparison of five landmarks located on the bone surface after the superimposition of T2 and Tv (ΔT2) (**A**) and T1 and Tv (ΔT1) (**B**) in patient 4 of the study group.

**Table 1 bioengineering-10-00914-t001:** Patient characteristics and surgery descriptions.

Group	Patient No.	Age	Gender	Operation	Diagnosis
	Pt 1	29	M	2jaw	III
	Pt 2	23	M	2jaw/gen	III
	Pt 3	29	F	1jaw/gen	III, FA
	Pt 4	20	F	2jaw	III, FA
Control	Pt 5	21	M	2jaw	III, FA
	Pt 6	18	F	2jaw	III
	Pt 7	26	M	2jaw/gen	III, FA
	Pt 8	22	F	2jaw	FA
	Pt 9	22	F	2jaw/gen	III, FA
	Pt 10	27	F	2jaw	III, FA
	Pt 11	19	F	2jaw/gen	III, FA
	Pt 1	26	M	2jaw	III, FA
	Pt 2	20	F	1jaw	III, FA
	Pt 3	22	M	2jaw	FA
	Pt 4	20	F	1jaw	III, FA
Study	Pt 5	22	M	1jaw	III, FA
	Pt 6	18	F	2jaw	III, FA
	Pt 7	32	M	1jaw	III, FA
	Pt 8	19	F	1jaw/gen	III, FA
	Pt 9	21	M	2jaw	FA
	Pt 10	21	F	2jaw	III, FA
	Pt 11	21	M	1jaw/gen	III, FA

2jaw: two jaw surgery; 1jaw: mandibular surgery; gen: genioplasty; III: skeletal class III malocclusion; FA: facial asymmetry.

**Table 2 bioengineering-10-00914-t002:** Descriptive characteristics of the participants.

		Group		Total	χ^2^	*p*-Value
		Control	Study			
Age		23.27 ± 1.18	22 ± 1.18		6.333	0.706
Gender	Male	4 (36.4)	6 (54.5)	10 (45.5)	0.733	0.392
	Female	7 (63.6)	5 (45.5)	12 (54.5)		
1jaw/2jaw	1jaw	1 (9.1)	6 (54.5)	7 (31.8)	5.238	0.022
	2jaw	10 (90.9)	5 (45.5)	15 (68.2)		
FA	FA (−)	3 (27.3)	0 (0)	3 (13.6)	3.474	0.062
	FA ( + )	8 (72.7)	11 (100)	19 (86.4)		
Total		11 (100)	11 (100)	22 (100)		

**Table 3 bioengineering-10-00914-t003:** Changes in the condyle-fossa distance by group, fossa location, and periods.

Source	Sum of Squares (SS)	df	Mean Squares (MS)	F	*p*-Value
Time	0.442	1	0.442	4.004	0.048
Time × group	0.015	1	0.015	0.137	0.712
Time × position of joint space	0.186	2	0.093	0.841	0.434
Error	13.905	126	0.110		

**Table 4 bioengineering-10-00914-t004:** Differences in distance over time by the position of the joint space for each group.

Group	Position of Joint Space	Time	N	Average (mm)	Standard Error	t	*p*-Value
Control	AJS	T0	22	1.95	0.70	−1.063	0.294
		T2	22	2.16	0.63		
	SJS	T0	22	2.40	1.01	−0.305	0.762
		T2	22	2.49	0.96		
	PJS	T0	22	2.15	0.88	0.058	0.954
		T2	22	2.13	0.67		
Study	AJS	T0	22	1.80	0.71	−0.504	0.617
		T2	22	1.90	0.60		
	SJS	T0	22	2.31	1.18	0	1
		T2	22	2.31	0.90		
	PJS	T0	22	2.02	0.92	−0.412	0.683
		T2	22	2.12	0.68		

**Table 5 bioengineering-10-00914-t005:** Differences based on group, point, and time.

Source		Sum of Squares	df	MEAN Squares	F	*p*-Value
Group	X diff.	0.009	1	0.009	0.165	0.685
	Y diff.	0.029	1	0.029	1.374	0.243
	Z diff.	0.003	1	0.003	0.087	0.768
	Surface diff.	0.039	1	0.039	0.449	0.503
Point	X diff.	0.38	4	0.095	1.813	0.128
	Y diff.	0.226	4	0.056	2.678	0.033
	Z diff.	0.295	4	0.074	1.851	0.121
	Surface diff.	0.795	4	0.199	2.262	0.064
Time	X diff.	0.001	1	0.001	0.019	0.891
	Y diff.	0.046	1	0.046	2.192	0.14
	Z diff.	0.937	1	0.937	23.525 **	0
	Surface diff.	0.559	1	0.559	6.365	0.012
Group × point	X diff.	0.109	4	0.027	0.518	0.722
	Y diff.	0.011	4	0.003	0.126	0.973
	Z diff.	0.063	4	0.016	0.394	0.813
	Surface	0.096	4	0.024	0.273	0.895
Group × time	X diff.	0.018	1	0.018	0.34	0.561
	Y diff.	0.033	1	0.033	1.542	0.216
	Z diff.	0.046	1	0.046	1.143	0.286
	Surface diff.	0.071	1	0.071	0.81	0.369
Point × time	X diff.	1.591	4	0.398	7.588 **	0
	Y diff.	0.174	4	0.043	2.061	0.087
	Z diff.	8.279	4	2.07	51.965 **	0
	Surface diff.	11.646	4	2.911	33.149 **	0
Group × point × time	X diff.	0.104	4	0.026	0.496	0.739
	Y diff.	0.03	4	0.008	0.357	0.839
	Z diff.	0.104	4	0.026	0.655	0.624
	Surface diff.	0.253	4	0.063	0.72	0.58
Error	X diff.	10.484	200	0.052		
	Y diff.	4.217	200	0.021		
	Z diff.	7.966	200	0.04		
	Surface diff.	17.566	200	0.088		

** *p* < 0.01.

**Table 6 bioengineering-10-00914-t006:** Differences in the time difference for each group.

	Group	Time diff.	Average (mm)	Standard Error	N
X diff.	Control	ΔT2	0.017	0.222	55
		ΔT1	0.004	0.226	55
	Study	ΔT2	0.012	0.271	55
		ΔT1	0.034	0.247	55
Y diff.	Control	ΔT2	0.035	0.116	55
		ΔT1	0.030	0.140	55
	Study	ΔT2	0.082	0.179	55
		ΔT1	0.028	0.146	55
Z diff.	Control	ΔT2	−0.070	0.233	55
		ΔT1	0.089	0.328	55
	Study	ΔT2	−0.059	0.242	55
		ΔT1	0.071	0.309	55
surface diff.	Control	ΔT2	0.049	0.308	55
		ΔT1	−0.088	0.443	55
	Study	ΔT2	0.039	0.335	55
		ΔT1	−0.025	0.399	55

**Table 7 bioengineering-10-00914-t007:** Differences by the time difference for each group and point within the group.

	Group	Point	Time diff.	Average (mm)	Standard Error (SE)	N
X diff.	Control	Lt. mental fo	ΔT2	0.08	0.17	11
			ΔT1	−0.15	0.15	11
		Lt. molar root	ΔT2	−0.01	0.22	11
			ΔT1	0.09	0.32	11
		Rt. mental fo	ΔT2	−0.06	0.20	11
			ΔT1	0.13	0.20	11
		Rt. molar root	ΔT2	0.03	0.33	11
			ΔT1	−0.13	0.13	11
		Incisor root	ΔT2	0.06	0.16	11
			ΔT1	0.07	0.14	11
	Study	Lt. mental fo	ΔT2	0.13	0.17	11
			ΔT1	−0.16	0.13	11
		Lt. molar root	ΔT2	−0.14	0.31	11
			ΔT1	0.08	0.32	11
		Rt. mental fo	ΔT2	−0.06	0.20	11
			ΔT1	0.15	0.19	11
		Rt. molar root	ΔT2	0.01	0.34	11
			ΔT1	0.00	0.30	11
		Incisor root	ΔT2	0.12	0.23	11
			ΔT1	0.09	0.17	11
Y diff.	Control	Lt. mental fo	ΔT2	0.00	0.02	11
			ΔT1	−0.02	0.06	11
		Lt. molar root	ΔT2	0.02	0.10	11
			ΔT1	−0.02	0.20	11
		Rt. mental fo	ΔT2	0.04	0.06	11
			ΔT1	0.06	0.10	11
		Rt. molar root	ΔT2	0.01	0.19	11
			ΔT1	0.08	0.10	11
		Incisor root	ΔT2	0.10	0.13	11
			ΔT1	0.04	0.19	11
	Study	Lt. mental fo	ΔT2	0.00	0.08	11
			ΔT1	0.00	0.05	11
		Lt. molar root	ΔT2	0.06	0.22	11
			ΔT1	−0.01	0.20	11
		Rt. mental fo	ΔT2	0.12	0.14	11
			ΔT1	0.08	0.17	11
		Rt. molar root	ΔT2	0.06	0.21	11
			ΔT1	0.08	0.08	11
		Incisor root	ΔT2	0.17	0.19	11
			ΔT1	−0.01	0.17	11
Z diff.	Control	Lt. mental fo	ΔT2	0.09	0.16	11
			ΔT1	−0.17	0.15	11
		Lt. molar root	ΔT2	−0.03	0.23	11
			ΔT1	0.09	0.30	11
		Rt. mental fo	ΔT2	0.04	0.16	11
			ΔT1	−0.11	0.16	11
		Rt. molar root	ΔT2	−0.09	0.14	11
			ΔT1	0.08	0.13	11
		Incisor root	ΔT2	−0.46	0.39	11
			ΔT1	0.72	0.48	11
	Study	Lt. mental fo	ΔT2	0.16	0.17	11
			ΔT1	−0.16	0.13	11
		Lt. molar root	ΔT2	−0.11	0.23	11
			ΔT1	0.08	0.30	11
		Rt. mental fo	ΔT2	0.04	0.16	11
			ΔT1	−0.12	0.15	11
		Rt. molar root	ΔT2	−0.01	0.24	11
			ΔT1	0.05	0.16	11
		Incisor root	ΔT2	−0.41	0.40	11
			ΔT1	0.47	0.38	11
surface diff.	Control	Lt. mental fo	ΔT2	−0.12	0.23	11
			ΔT1	0.23	0.22	11
		Lt. molar root	ΔT2	0.01	0.29	11
			ΔT1	0.00	0.38	11
		Rt. mental fo	ΔT2	−0.08	0.26	11
			ΔT1	0.17	0.25	11
		Rt. molar root	ΔT2	0.06	0.34	11
			ΔT1	−0.17	0.20	11
		Incisor root	ΔT2	0.37	0.16	11
			ΔT1	−0.72	0.48	11
	Study	Lt. mental fo	ΔT2	−0.21	0.24	11
			ΔT1	0.23	0.19	11
		Lt. molar root	ΔT2	0.13	0.30	11
			ΔT1	−0.05	0.44	11
		Rt. mental fo	ΔT2	−0.08	0.26	11
			ΔT1	0.20	0.24	11
		Rt. molar root	ΔT2	0.05	0.46	11
			ΔT1	−0.03	0.36	11
		Incisor root	ΔT2	0.33	0.17	11
			ΔT1	−0.50	0.38	11

diff.: difference, Rt.: right, Lt.: left, fo: foramen.

## Data Availability

The datasets generated during and/or analyzed during the current study are available from the corresponding author upon reasonable request.

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
