# Peer review of "Clinical Stability of Bespoke Snowman Plates for Fixation following Sagittal Split Ramus Osteotomy of the Mandible"

_bioengineering, 2023, doi:10.3390/bioengineering10080914_

Round 1

Reviewer 1 Report

This study is clinical stability of bespoke Snowman plates for fixation following sagittal split ramus osteotomy of the mandible

The manuscript is well written. However, it can be further improved.

There are only few comments here:

Page 4, Figure 2 and Figure 3 show panoramic views of pre- and post-operative images. It was never mentioned anywhere in the manuscript, especially in the Materials and Methods section. Only CBCT was mentioned in page 3 line 119 - Cone-beam computed tomography (CBCT) scans were performed before surgery (T0), four months after surgery 120 (T1), and one year after surgery (T2). The virtual surgery image is named T virtual (Tv).

Page 5, line 154. The CBCT images were converted to the Digital Imaging and Communications in 154 Medicine (DICOM) format. This is a wrong statement. DICOM is the standard format protocol for CBCT.

Page 5, line 172. Which materials were printed using Creality 3D printer and which materials printed using 5-axis CNC milling machine?

Page 11, line 273. Why p values < 0.01 was chosen instead of <0.05 to be considered statistically significant?

Page 16, caption of Figure 12. Comparison of five landmarks located on the bone surface after the superposition of T2 and Tv (ΔT2) (A) and T1 and Tv (ΔT1 )(B) in patient 4 of the control group. Do you mean superimposition?

Page 17, caption of Figure 13. Comparison of five landmarks located on the bone surface after the superposition of T2 and Tv (ΔT2) (A) and T1 and Tv (ΔT1) (B) in patient 4 of the study group. Do you mean superimposition?

Moderate editing is required

Reviewer 2 Report

The present article follows a very relevant topic, with a lot of clinical relevance. Fragment fixation and exact correlation of the preoperative planning and the actual results after orthognathic surgery is clearly the most difficult aspect of the entire orthognathic treatment. Some aspects must be taken into account before continuing with the publication of this paper:  

L66: Consequently, the stability of the mandible after surgery is more greatly dependent on the method of bone fixation than that of the maxilla - this depends on the surgical approach - for example, in maxilla first surgeries, this statement is not correct. 

The type of study is not clearly mentioned.

The osteotomy of the mandible is not clearly described. Please describe the osteotomy lines, especially on the internal part of the ramus. Also, more information should be provided on the whole surgical protocol.

The superimposition of the scans and planning are very eloquent from a visual point of view. Also, comparing certain points is very relevant. But, would it be possible to obtain a percentage of superimposition? Because that would be most relevant.

For more accurate results, patients with different diagnosis should not be included in the same study. Only class III patients, given the fact that these are the simplest surgeries should be perhaps included. Also, monomaxillary and bimaxillary surgeries should not be combined in the same study. 

The most important aspect, the intraoperative condyle repositioning technique is not described. Please insist on this and also on the inconveniences that might arise from an incorrect positioning, both intraoperative and especially postoperative. 
